# Human Motor Noise Assessed by Electromagnetic Sensors and Its Relationship with the Degrees of Freedom Involved in Movement Control

**DOI:** 10.3390/s23042256

**Published:** 2023-02-17

**Authors:** Carla Caballero, David Barbado, Francisco J. Moreno

**Affiliations:** 1Sport Sciences Department, Miguel Hernandez University of Elche, 03202 Alicante, Spain; 2Alicante Institute for Health and Biomedical Research (ISABIAL), 03010 Alicante, Spain

**Keywords:** motor variability, nonlinear tools, degrees of freedom, kinematic, tracking, electromagnetic sensor

## Abstract

Motor variability is a prominent feature of the human movement that, nowadays, can be easily measured through different sensors and analyzed using different types of variables, and it seems to be related to functional and adaptative motor behavior. It has been stated that motor variability is related to the system’s flexibility needed to choose the right degrees of freedom (DoFs) to adapt to constant environmental changes. However, the potential relationship between motor variability and DoFs is unknown. The aim of this study was to analyze how motor variability, both the amount and structure, changes depending on the mechanical DoFs involved in the movement control. For this purpose, movement variability was assessed by a tracking sensor in five tasks with different DoFs, and the amount, using standard deviation, and the structure of variability, through fuzzy entropy and detrended fluctuation analysis, were also assessed. The results showed a higher amount of variability and a less predictable and more auto-correlated variability structure in the long-term when more mechanical DoFs are implied. The studies that analyze motor variability should consider the type of movement and the DoFs involved in the analyzed task since, as the findings have shown, both factors have a noticeable influence on the amount and the structure of motor variability.

## 1. Introduction

Human movements are the result of the complex interaction between all the independent elements that make up the human body and their relationship with the environment [1]. The aim of this interaction is to achieve a successful motor solution by choosing a suitable degree of freedom (DoF) configuration among the multiple and redundant DoFs depending on the performer’s level and environment inputs [2], i.e., to choose the functional DoFs (defined as the very limited DoF during specific movement tasks) from all the available mechanical DoFs (defined as the minimum number of independent coordinates needed to describe a system’s position) [3]. Different authors have backed the idea that motor variability gives the system the flexibility to choose the right DoFs to facilitate the adaptation to environmental changes [4,5]. How does this work? Human motor variability would promote these adaptive behaviors by facilitating the exploration of the motor system redundancy [6] and the ability to perform motor adjustments [7] in order to refine movement performance during the interaction with changing environments. Thus, the study of motor system variations has been an interesting topic for several researchers to study how variability changes according to the skill level or the learning stages [8,9,10] and to search for the optimal level of variability to optimize the adaptation process [4]. 

Although there may not be a single pathway of change in the evolving patterns of coordination, Bernstein [8] suggested that “freezing” DoF strategy would simplify the motor control problem, with the intention of decreasing motor variability. More efficient solutions to the DoF problem would involve gradual unfreezing or “freeing” of DoFs. Nevertheless, this motor control strategy can paradoxically imply an increase in motor variance in joint coupling but a reduced output variability. Researchers are still investigating how movement synergies actually simplify the problem of movement coordination [8,11,12,13]. To the authors’ knowledge, how motor variability changes according to the DoFs chosen for a specific motor task is still unknown. 

It must be taken into account to fully understand the potential relationship between motor variability and the DOFs that the movement dynamics of the biological systems is nonlinear [14,15,16]. Thus, even though the amount of variability is deeply related to the exploration of the individual to find a suitable solution [6,17], it does not inform about how movement behavior changes over time [18]. Nonlinear tools allow the acquisition of information about the variability structure, which allows the quantification of the dynamics of the system’s behavior [4,19,20,21]. In fact, as it was mentioned above, the amount of variability has already related to the freezing/unfreezing DoF process [2]. However, the analysis of motor variability structure according the mechanical DoFs involved in movement controls has not been deeply investigated. Some previous studies about postural control suggested that there could be a relationship between the variability structure and how the individual coordinates the DoFs to accomplish a required task, proposing the complementary use of nonlinear analysis to contrast the hypothesis of freezing and freeing DoFs [22].

Thus, the main aim of this study was to analyze how motor variability changes depending on the mechanical DoFs involved in movement control. It was hypothesized that the observable motor behavior of the hand (endpoint) will be increasingly driven by the arm kinematic demands when a release of the DoFs occurs, resulting in more predictable and auto-correlated motor behavior. In order to perform that, the variability structure of the movement kinematic was assessed through fuzzy entropy (FuzzyEn) and detrended fluctuation analysis (DFA), which provided information about the predictability and autocorrelation of the signal, respectively. As a secondary aim, the behavior of the different body segments was compared during the upright stance to understand how joint variability output changes depending on whether the body segments are proximal or distal to the force application point (ground forces).

## 2. Materials and Methods

### 2.1. Participants

Twenty-one healthy volunteers (9 females and 12 males), with no neuromuscular pathology, took part in this study (age = 30.6 ± 8.28 years).

Written informed consent was obtained from each participant prior to testing. Data were treated anonymously, and all participants were informed of the risks and benefits of the trial. The study was conducted according to the guidelines of the Declaration of Helsinki and was approved by the University Office for Research Ethics (DPS.FMH.01.13).

### 2.2. Instruments and Procedure

Participants performed five different simple tasks in one session (Figure 1), in which the kinematic behavior of the distal segment, the arm, was manipulated, from static and constraint conditions to dynamic tasks. Specifically, participants performed three static and two dynamic situations. The static situations consisted of: (a) placing the dominant hand on a stable surface (HoS); (b) keeping the arm extended down by the participant’s side in a relaxed position (ARelax); (c) keeping their arm extended in front of them, with the shoulder at 90° with respect to the vertical axis (A90). The dynamic situations consisted of: (d) from the c situation, participants performed oscillatory movements up and down in a range of 30 cm using a line placed in front of them (UpDown) as a reference; (e) from the c situation, participants performed circular movements in the same range following a circle (30 cm diameter) placed in front of them (Circle). Participants were asked to be as relaxed and as still as possible in all the static situations. Each task was maintained for 20 s and repeated 3 times with 1′ rest between trials. The different tasks were counterbalanced between participants.

An electromagnetic tracking sensor (Polhemus Liberty^®^, Polhemus, Colchester, VT, USA) was placed on the back of the participant’s hand to analyze the motor variability displayed in each task situation. In addition, in the static situation b (ARelax), two extra sensors were placed on the acromion (Shoulder) and on the anterior superior iliac spine (Hip). All sensors were placed on each participant’s dominant side, with the purpose of measuring the movement fluctuations of the different body places in which joints with different DoF are involved. Thus, a total of seven measurements were registered.

The Polhemus Liberty^®^ electromagnetic tracking system allowed us to measure the position in the three axes (anteroposterior (AP), mediolateral (ML), and vertical (V)) with an accuracy of 0.076 cm in the position and 0.15° in the angular orientation. The sampling frequency was 240 Hz.

### 2.3. Data Analysis and Reduction

A custom software program in Labview, 2009 (National Instruments, Austin, TX, USA) was used for data analysis. Kinematic time series were down-sampled from 240 to 100 Hz to avoid artificial collinearities caused by signal oversampling, which affect the data variability [23]. Data were already filtered by the Polhemus Liberty^®^ tracking system with a single-pole low-pass filter with an adaptive pole location. The pre-set filtering parameters were: sensitivity = 0.02; boundary (FLow) = 0.02; boundary (FHigh) = 0.8; max transition rate = 0.95. In addition, a high-pass filter (1 Hz cutoff frequency) was used to remove the low frequency fluctuations caused by the movement performed during the dynamic tasks. The time series length was 2000 data points.

All the variability measures were calculated over the resultant distance (*RD*) kinematic time series, instead of the AP, ML, and V axes, in order to obtain a global variable of the sensor displacement. The magnitude of the three axes of the sensors position was used to obtain a general index of the sensor displacement.
$$RD\;time\;series=\sqrt{{\left(x\left[n\right]-\bar{x}\right)}^{2}+{\left(y\left[n\right]-\bar{y}\right)}^{2}+{\left(z\left[n\right]-\bar{z}\right)}^{2}\;}\;\;\;\;\;\;n=1,2,\;\ldots ,\;N.$$

In which *n* is the number of data points in the kinematic time series. *X*, *Y*, and *Z* correspond to values of the kinematic time series for the AP, ML, and V axes, respectively, and $\bar{X},$ $\bar{Y},$ and $\bar{Z}$ correspond to the means of the kinematic time series for the AP, ML, and V axes, respectively.

The standard deviation (SD) of the sensor displacement was used to assess the variability amount displayed by the individuals in each situation. In order to assess the structure of the variability, fuzzy entropy (FE) [24] and detrended fluctuation analysis (DFA) [25] were used. FE measures the degree of regularity of the signal output by quantifying how short-term patterns are repeated over time. It computes the repeatability of vectors of length m and m + 1 that repeat within a tolerance range of r within the standard deviation of the time series. Higher values of FE indicate that vectors of length are less repeatable than vectors of length m + 1, highlighting the lower predictability of future data points and a greater irregularity within the time series. Lower values represent a greater repeatability of vectors of length m + 1, being a marker of higher regularity in the time series. For FE, the following parameter values were used: vector length, m = 2; tolerance window, r = 0.2 ∗ SD; and gradient, *n* = 2. These parameter values are the most frequently used because they show high consistency [24,26,27,28]. 

On the other hand, DFA evaluates the presence of long-term correlations within the time series by a parameter referred to as the scaling index α [25,29], identifying the extent to which further data are dependent on previous data [30]. Different values of α indicate the following: α > 0.5 implies persistence (i.e., the trajectory tends to continue in its current direction); α < 0.5 implies anti-persistence (i.e., the trajectory tends to return to where it came from); and α = 0.5 implies uncorrelated signal [29]. Its computation is based on the random walk theory that represents a modification of a classic root mean square analysis of the random walk. To maximize the long-range correlations and to reduce the estimation error of α, the long-term correlation was characterized by the slope obtained from the range of 4 ≤ *n* ≥ N/10, in which N is the data length [31].

### 2.4. Statistical Analysis

The normality of variable distribution was evaluated using the Kolmogorov-Smirnov test with the Lilliefors correction. A repeated measure ANOVA was applied to assess the effect of the movement implication (comparing HoS, ARelax, A90, UpDown, and Circle situations). In addition, another repeated measure ANOVA was used to assess the effect of how proximal or distal the joint is to the force application point (comparing the Hip, Shoulder, and Hand in the ARelax situation). In addition, other Alpha levels were set at *p* < 0.05.

## 3. Results

Average values obtained in each situation for each variable are displayed in Table 1.

First of all, significant differences were also found in all the variables when the variability of the hand displacement in the different situations with different movement implications were compared (SD: F4,80 = 15.156, *p* < 0.001, ƞp2 =0.431; FE: F4,80= 404.382, *p* < 0.001, ƞp2 = 0.953; DFA: F4,80 = 67.217, *p* < 0.001, ƞp2 = 0.771). As the DoFs and voluntary movement involved increased, the amount of variability increased, and the variability structure was more irregular and auto-correlated. Specifically, in the situations with voluntary movement, participants also showed significantly higher SD and lower FE values than in relaxed situations (HoS and ARelax situations). In addition, in the situations in which the muscular contractions were dynamic (UpDown and Circle), the participants presented significantly higher SD and lower FE values. No differences were found between UpDown and Circle situations. Regarding DFA, participants exhibited a significantly higher auto-correlation in the situations which involved voluntary movement (UpDown and Circle) than in the situations with no movement. There were significant differences between the situation of ARelax and A90, the latter showing lower auto-correlations. However, no significant differences were found between HoS and ARelax, nor between HoS and A90 (Figure 2).

In addition, significant differences were found between the sensors in all the variables (SD: F2,44 = 19.438, *p* < 0.001, ηp2 = 0.469; FE: F2,44 = 20.475, *p* < 0.001, ηp2 = 0.482; DFA: F2,44 = 16.331, *p* < *0*.001, ηp2 = 0.426) when comparing the variability of the sensors placed on the different body places in the ARelax situation. The amount of variability was significantly lower in the hip sensor than in the shoulder and hand sensor (*p* < 0.001), and its structure of variability was more irregular and auto-correlated (Figure 3).

## 4. Discussion

Measuring movement variability is a relevant research topic because it seems to provide useful information about motor behavior. Nowadays, how motor variability impacts motor performance and learning can be easily measured through different sensors and using several mathematical tools, including non-linear tools, which provide complementary information about the movement dynamic. In this sense, recent findings have shown how motor variability promotes better motor performance and a faster learning rate, which has been linked with a high system flexibility to manipulate the system’s DoFs properly to adapt to constant environmental changes [32,33]. However, no studies have assessed how the structure of motor variability changes according to the degrees of freedom involved in a given task. The main findings confirmed that the amount and the structure of the hand movement variability changes as the DoFs involved in motor tasks change.

Focusing on the amount of variability of the hand motion, participants increased their motor variability when the arm mechanical DOFs were increased, especially when they were asked to maintain their hand in front of them and move the hand in a repetitive cycle. Previous studies have related the increment of motor variability to a freeing strategy in DoF management, and it has been linked to an exploration strategy in which several DoFs are used to find the best synergy to reach the task goal [2,8]. In the present study, what was assessed is how the measurement of motor variability can be affected by the number of DoFs involved in different joints, and the results indicated that the amount of variability increased as a greater number of mechanical DoFs were involved. It suggests that the individual shows more motor variability as a higher number of elements are implied in self-organizing movements aimed to maintain a suitable motor solution. 

Regarding the structure of variability, when the participants were asked to maintain their hand in front of them and perform oscillatory movements up and down or in two-dimensional circular movement, a more predictable (low entropy) and auto-correlated (high long range DFA) variability structure was found, confirming our hypothesis. These results can be interpreted as opposite to previous studies that suggested that a system with a higher number of elements (DoFs) would exhibit a more complex behavior related to more functional behavior [7,34,35]. Kodama et al. [36] found a decrease in entropy (Sample Entropy) and an increase in DFA when the DoFs were constrained, while different trends according to the nonlinear tool used were found. It must be pointed out that the experimental protocols used to assess motor variability according to the number of DoFs involved were quite different to Kodama et al. [36]. They used the center of pressure to assess the variability in various balance tasks in which the DoFs were constrained by fixing some of the joints. 

Therefore, it has to be considered that the motor behavior dynamic is not just a question of the number of DoFs implied but how they are constrained. In the current study, the DoFs were freed along the tasks, but some constraints were also added, such as performing the movement in a specific space. This requirement could have caused a more restrictive motor behavior even when a higher number of DoFs are being used [37]. In this case, our findings would support other similar results found in a postural control study, in which participants had to adjust their movement to a moving target [22]. Participants displayed less autocorrelated COP fluctuations in the axis that was constrained by the movement of the target than in the one that was not. All these findings together reinforce the idea that the higher the prevalence of a task constraint, the higher the complexity of the motor output, which is reflected in lower autocorrelated fluctuations of the center of pressure time series data [7,38].

Finally, as a secondary aim, we explored how movement variability changes depending on whether the joint is proximal or distal to the force application point (ground forces) during an upright stance. Specifically, when we analyzed the ARelax condition, the more distal the segment was from the force application point (arm > shoulder > hip), the more the amount of variability was found. These findings are related to the fact that, during the upright stance, the human body works as an inverted pendulum [39]; thus, the sway of the distal segment is amplified even for the high movement frequencies analyzed in this study.

Regarding the structure of variability, when the segment measured was more distal to the force application point (ground forces), significantly lower long range DFA values were found compared to proximal segments (hip and shoulder). Interestingly, the entropy measure revealed the same result trend, but it was expected that it would follow the opposite trend, i.e., the less long-term dependency of the medium and high frequency oscillations of the distal segment (quantified by the DFA) was associated with increased repeatability of the short-term patterns over time (quantified by the fuzzy entropy). From the authors’ point of view, although there is no clear explanation of these apparently contradictory findings, they seem to reinforce that entropy and autocorrelation measures do not provide the same information about the dynamic of the variability structure [18,40].

### Limitations and Final Remarks

The changes in the features of motor variability probably have multifactorial causes that are very difficult to isolate. A limitation of this study is that several features were modified along the different experimental conditions, not only the amount of DoFs and movement involved but also the constraints used. For example, there are some task conditions in which the hand of the participant would be still and relaxed, while in other conditions the hand is moving. The intention the individual has during the task performed is another factor that seems to affect motor variability characteristics [37,41]. In addition, the type of muscular contraction is also an aspect to consider. To the authors’ knowledge, there are no studies that compare different types of dynamic and isometric contractions, but the literature shows that there is a decrease in complexity (reflected by lower entropy or higher DFA) when fatigue protocols using isometric contractions are used [42,43]. However, this trend is not so clear when dynamic contractions are used. Using this type of contraction, some authors have also found a decrease in complexity [44], but other studies have found that complexity increases after the fatigue protocol [45,46], or it depends on the variables used to assess the structure of variability [47]. In this study, isometric (maintaining the hand with the arm tight in front of the body) and dynamic contractions (moving the hand up-down or in a circle) were compared, and the highest differences, not only in the amount of variability but in its structure were found between the conditions in which there was no movement and the ones in which movement was required, with these last conditions being more variable, predictable, and autocorrelated. In addition, even between situations with no movement, significant differences were found in the amount of variability and DFA values, finding lower variability magnitude and higher autocorrelation when the hand was relaxed (ARelax) than when the hand was kept still in front of the body, with the shoulder at 90° with respect to the vertical axis in a 90° angle (A90). It seems that the higher co-contraction maintained to keep the hand as still as possible reduced autocorrelation. Differences according to joint stiffness were previously found to be related to the level of freezing DoFs [48,49], but more research on the effect the muscle contraction has on DoF configuration and motor variability evolution would be necessary. Finally, the fact that no differences were found between these two situations using FE indicates that DFA is more sensitive to the features of the muscular tone. Therefore, the use of complementary nonlinear measures to analyze the dynamics of motor variability is recommended.

## Figures and Tables

**Figure 1 sensors-23-02256-f001:**
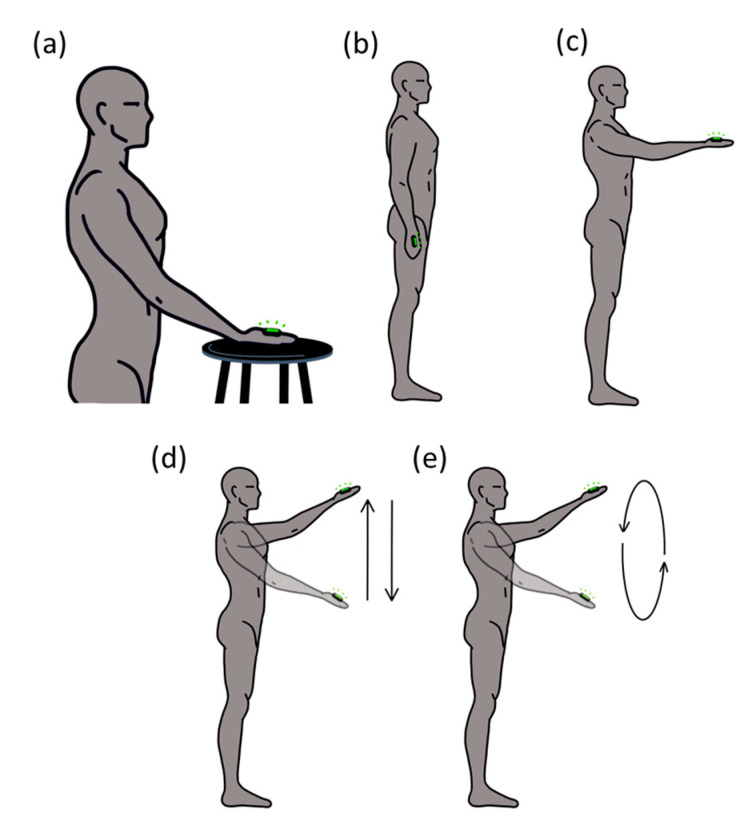
The sequence of the five different tasks: (**a**) placing the dominant hand on a stable surface (HoS); (**b**) keeping the arm extended down by the participant’s side in a relaxed position (ARelax); (**c**) keeping their arm extended in front of them, with the shoulder at 90° with respect to the vertical axis (A90); (**d**) from the c situation, participants performed oscillatory movements up and down in a range of 30 cm using a line placed in front of them (UpDown) as a reference; (**e**) from the c situation, participants performed circular movements following a circle (30 cm diameter) placed in front of them (Circle).

**Figure 2 sensors-23-02256-f002:**
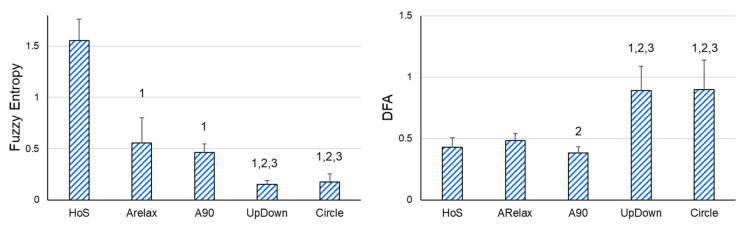
Pairwise Comparisons between the different situations. DFA = Detrended Fluctuation Analysis; HoS = placing the dominant hand on a stable surface situation; ARelax = keeping the arm extended down by the participant’s side in a relaxed position; A90 = keeping their arm extended in front of them, with the shoulder at 90° with respect to the vertical axis situation; UpDown = from the A90 situation, the participants performed oscillatory movements up and down in a range of 30 cm using a line placed in front of them as a reference; Circle = from the A90 situation, participants performed circular movements following a circle (30 cm diameter) placed in front of them. 1 = significant differences compared to the HoS situation; 2 = significant differences compared to the ARelax situation; 3 = significant differences compared to the A90 situation.

**Figure 3 sensors-23-02256-f003:**
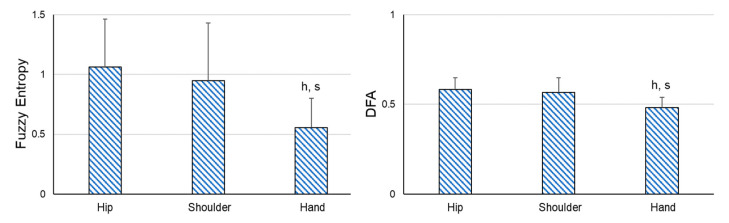
Pairwise comparisons between the different sensors in the static and relaxed situations. DFA = detrended fluctuation analysis. h = significant differences compared to the hip sensor; s = significant differences compared to the shoulder sensor. *p* ≤ 0.001.

**Table 1 sensors-23-02256-t001:** Mean ± SD values of the amount (standard deviation) and structure (fuzzy entropy and detrended fluctuation analysis) of motor variability in every situation.

	HoS	ARelax			A90	UpDown	Circle
		Hip	Shoulder	Hand			
SD	0.004 ± 0.002	0.009 ± 0.009	0.011 ± 0.008	0.027 ± 0.015	0.043 ± 0.007	1.166 ± 1.317	0.567 ± 0.550
FE	1.545 ± 0.207	1.064 ± 0.398	0.965 ± 0.483	0.544 ± 0.243	0.462 ± 0.085	0.152 ± 0.041	0.184 ± 0.081
DFA	0.427 ± 0.080	0.583 ± 0.064	0.569 ± 0.082	0.486 ± 0.057	0.387 ± 0.042	0.875 ± 0.190	0.880 ± 0.244

SD = standard deviation; FE = fuzzy entropy; DFA = detrended fluctuation analysis. HoS = placing the dominant hand on a stable surface situation; ARelax = keeping the arm extended down by the participant’s side in a relaxed position; A90 = keeping the arm extended in front of them, with the shoulder at 90° with respect to the vertical axis situation; UpDown = from the A90 situation, the participants performed oscillatory movements up and down in a range of 30 cm using a line placed in front of them as a reference; Circle = from the A90 situation, participants performed circular movements following a circle (30 cm diameter) placed in front of them; 1 = significant differences compared to the HoS situation; 2 = significant differences compared to the ARelax situation; 3 = significant differences compared to the A90 situation.

## Data Availability

The data presented in this study are available at https://doi.org/10.6084/m9.figshare.22100528 (accessed on 21 December 2022).

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
