# Peer review of "Human Motor Noise Assessed by Electromagnetic Sensors and Its Relationship with the Degrees of Freedom Involved in Movement Control"

_sensors, 2023, doi:10.3390/s23042256_

Round 1

Reviewer 1 Report

The manuscript investigates how different joints vary differently and how different tasks (static, dynamic) affect hand position variability. Despite my large interest on the topic of variability, the introduction and some methodological choices must be further developed.

Major comments:

- Variability is under defined up to the methods section. Motor variability has different "facets" (dispersion [e.g., SD, in terms of the goal space], structure [e.g., short-term, long-term, specific dynamics]) that have different "influences" on motor behavior. The authors do not specify what is the "facet" they want to study. For instance, the discussion on motor variability helping the system to find the most appropriate solution is largely related to dispersion (and can be differentiated into "falling into more stable solutions" Schöner et al., 1992, JMB, 24(1); or finding the solution of a task Pacheco et al., 2017, Ec. Psych). Nonetheless, irregularity and auto-correlation is also discussed in the paper. My suggestion is for the authors to be more specific on what they want to mean in terms of variability and how they want to study that. If the question is about all facets, all facets of motor variability must be discussed properly (with relevant literature on it).

- The second concept mainly discussed, degrees of freedom, is also poorly motivated. What are the main DoFs being discussed dynamical, biomechanical? How do they relate to the conditions? It seems that the authors assume that the hip "involves" less DoFs while the hands involve more. What is to "involve" more DoFs? The authors also make use of literature on learning (and DoFs) while their paradigm do not relate to learning at all. I suggest the authors to extend their description of what is meant and how do DoFs would, theoretically, relate to variability.

- I am not sure whether the main manipulation holds. The main goal of the paper is "to analyze how motor variability changes depending on the mechanical DoFs involved". The manipulation is performed within the same condition. Nonetheless, this ignores the fact that, under the same condition, different joints compensate one for another - not much in terms of the number of DoFs but rather in terms of the task being performed. In the task discussed (ARelax condition, a standing task), the hand has no goal to achieve while the hip is closer to the center of mass.

See: Riley, M. A., Stoffregen, T. A., Grocki, M. J., & Turvey, M. T. (1999). Postural stabilization for the control of touching. Human Movement Science, 18, 795–817; Stoffregen, T. A., Pagulayan, R. J., Bardy, B. G., & Hettinger, L. J. (2000). Modulating postural control to facilitate visual performance. Human Movement Science, 19(2); Stoffregen, T. A., Smart, L. J., Bardy, B. G., & Pagulayan, R. J. (1999). Postural stabilization of looking. JEP: Human Perception and Performance, 25(6), 1641–1658; Lee, Pacheco, & Newell. 2019 Human Movement Science 66, 425-439.

Minor points:

- Abstract:

Line 18 - the authors should make it clear how a signal can be less predictable while being more auto-correlated (they should add the term "long-term")

- Introduction:

General - The authors make strong affirmations in terms of "optimal configuration" (Line 29), "very limited DOF" (line 30), "human system always looks for the most stable solution" (lines 35-36), and "motor variability helps the system to find the most successful and stable..." (lines 38-39) (for instance) without much support from the literature. My suggestion is to either cite the ones demonstrating it, or consider the appropriate terms. Considering the last one, I would be careful in citing Davids et al. (2006) as they are not the primary source of evidence backing up the affirmation made.

Second paragraph: Please unpack the meaning of variability. One can "free" DoFs while being consistent over trials or having a more consistent hand path (for instance).

Line 59 - It is a strong affirmation to claim the human movement system as chaotic. The references cited might not be sufficient for such a claim as, as far as I am aware, they do not demonstrate sensitivity to initial conditions.

Line 59 - "characterized by chaotic dynamics (12-14). That means that the motor variability..." The sentence seems to imply that because the system is chaotic it has deterministic and random parts. A system being chaotic does not mean that it has random and deterministic part. A chaotic system is deterministic by definition.

Lines 61-62 - Why is the random part diminished? Is the argument made in terms of Riley & Turvey or Newell & Corcos? They are not empirical papers demonstrating this.

Line 70 - "More auto-organized". If I am not mistaked, there is no such thing as more auto-organized (I believe the authors mean "self-organized). Self organization is categorical. Either a system is controlled by (central, external) entity, or the interaction of parts lead to organization (self-organization).

Methods:

- The tasks are poorly motivated.

- The introduction does not motivate the dynamics/static nature of the tasks being compared.

Lines 121-123: Why were the low frequency fluctuations discarded?

Line 139 - a letter "m" is missing in between "vectors of length" and "are less repeatable".

Results:

Line 171: The authors claim more irregularity and auto-correlation. Is it because were are talking about auto-correlation in the long range? That is why it is "contradictory" to irregularity? If this is the case, this must be made clear in a sentence in the methods.

Discussion:

Lines 218-221: The authors should be careful as Bernstein did not refer to synergies in the same sense as Latash puts it currently (in terms of redundant, non-redudant spaces). My suggestion is for the authors to specify clearly what is meant here.

Author Response

We thank the editor and the reviewers for the opportunity to enhance the quality of the manuscript. Below is each issue raised by the reviewers (in bold font) followed by the answers. The changes in the manuscript have been marked up using the Track Changes:

REVIEWER 1:

Major comments

  1. Variability is under defined up to the methods section. Motor variability has different "facets" (dispersion [e.g., SD, in terms of the goal space], structure [e.g., short-term, long-term, specific dynamics]) that have different "influences" on motor behavior. The authors do not specify what is the "facet" they want to study. For instance, the discussion on motor variability helping the system to find the most appropriate solution is largely related to dispersion (and can be differentiated into "falling into more stable solutions" Schöner et al., 1992, JMB, 24(1); or finding the solution of a task Pacheco et al., 2017, Ec. Psych). Nonetheless, irregularity and auto-correlation are also discussed in the paper. My suggestion is for the authors to be more specific on what they want to mean in terms of variability and how they want to study that. If the question is about all facets, all facets of motor variability must be discussed properly (with relevant literature on it).

Response:

The authors want to thank this comment. We agree with the author about all the facets variability has. In fact, in the manuscript, it has been differentiated the amount and the structure of variability. However, we understand that these differences have not been properly justified and discussed. Thus, the authors have added more information about these two different features of motor variability in the introduction of the paper.

Lines 63-80: It must be taken into account to fully understand the potential relationship between motor variability and the DOFs that the movement dynamics of the biological systems is nonlinear (14–16). Thus, even the amount of variability has been deeply related to the exploration of the individual to find a suitable solution (6,17), it does not inform about how movement behavior changes over time (18). Nonlinear tools allow the acquisition of information about the variability structure which allows the quantification of the dynamics of the system's behavior (4,19-21). In fact, as it has been mentioned above, the amount of variability has already related to the freezing/unfreezing DoFs process (2). However, the analysis of motor variability structure according the mechanical DoFs involved in movement controls has not been investigated in deep. Some previous studies about postural control have suggested that there could be a relationship between the variability structure and how the individual coordinated the DoFs to accomplish a required task, proposing the complementary use of nonlinear analysis to contrast the hypothesis of freezing and freeing DoF (22)”.

  1. The second concept mainly discussed, degrees of freedom, is also poorly motivated. What are the main DoFs being discussed dynamical, biomechanical? How do they relate to the conditions? It seems that the authors assume that the hip "involves" less DoFs while the hands involve more. What is to "involve" more DoFs? The authors also make use of literature on learning (and DoFs) while their paradigm does not relate to learning at all. I suggest the authors to extend their description of what is meant and how do DoFs would, theoretically, relate to variability.

Response:

Thank you very much for the observations. We appreciate this comment that helps us to improve the conceptualisation of our work. The reviewer is right regarding both suggestions. On the one hand, from a mechanical concept of the degree of freedom (Li, 2006), both the hip and hand (as well as the ankle) have the same amount of “independent coordinates to describe their position and achieve a performer´s goal”. This is, as rigid 3D bones/joints, their movements have six DOF: three translations and three rotations. So, technically, we only compared tasks with different DOF when we compared the hand variability under different conditions. Regarding the ARelax task, when we compared hand, shoulder and hip behaviour under the same condition, we used the term “DOF implication” to discriminate how proximal and distal joints/bones are controlled during a given task, but we are aware that this can be confusing. While at first sight the more distal position of the sensor, the higher number of joints implied in the resultant output movement, it is true that proximal segment variations are also affected by compensatory movements of distal segments in a synergic relationship. Therefore, when we compared the different joints in the ARelax task (upright stance), we aimed to analyse how joint variability output changes depending on whether the joint is proximal or distal to the force application point (ground forces). We have modified the presentation of the methods, the statistical analysis of the results, and the discussion section. Please, check the tracking changes in the manuscript reviewed. On the other hand, we have modified the introduction section according to the reviewer´s suggestion reducing the “learning” rationale. Conversely, we have now focused on how changes in DOFs are associated with motor variability.

Lines 49-62: “Although there may not be a single pathway of change in the evolving patterns of coordination, Bernstein (8) suggested that "freezing" DoFs strategy would simplify the motor control problem, with the intention of decreasing motor variability. More efficient solutions to the DoFs problem would involve gradual unfreezing or "freeing" of DoFs. Nevertheless, this motor control strategy can paradoxically imply an increase in motor variance in joints coupling but a reduced output variability. Researchers are still investigating how movement synergies actually simplify the problem of movement coordination (8,11-13). To the authors' knowledge, how motor variability changes according to the DOFs chosen for a specific motor task is still unknown.

  1. I am not sure whether the main manipulation holds. The main goal of the paper is "to analyze how motor variability changes depending on the mechanical DoFs involved". The manipulation is performed within the same condition. Nonetheless, this ignores the fact that, under the same condition, different joints compensate one for another - not much in terms of the number of DoFs but rather in terms of the task being performed. In the task discussed (ARelax condition, a standing task), the hand has no goal to achieve while the hip is closer to the center of mass.

See: Riley, M. A., Stoffregen, T. A., Grocki, M. J., & Turvey, M. T. (1999). Postural stabilization for the control of touching. Human Movement Science, 18, 795–817; Stoffregen, T. A., Pagulayan, R. J., Bardy, B. G., & Hettinger, L. J. (2000). Modulating postural control to facilitate visual performance. Human Movement Science, 19(2); Stoffregen, T. A., Smart, L. J., Bardy, B. G., & Pagulayan, R. J. (1999). Postural stabilization of looking. JEP: Human Perception and Performance, 25(6), 1641–1658; Lee, Pacheco, & Newell. 2019 Human Movement Science 66, 425-439.

Response:

As presented above, we have tried to clarify the experiment conditions that we have manipulated. On the one hand, the manipulations of the DOFs involved in controlling the different tasks’ goals have been done by only modifying the arm behaviour. On the other hand, as a secondary aim, we compared the behaviour of the different joints during the ARelax task to describe how joint variability output changes depending on whether the joint is proximal or distal to the force application point (ground forces). This information has been clarified in the introduction and method sections of the study.

Lines 81-93: “Thus, the main aim of this study was to analyze how motor variability changes depending on the mechanical DoFs involved in movement control. It was hypothesized that the observable motor behavior of the hand (endpoint) will be increasingly driven by the arm kinematic demands when a release of the DOFs occurs, resulting in more predictable and auto-correlated motor behavior. In order to do that, the variability structure of the movement kinematic was assessed through the FuzzyEntropy (FuzzyEn) and Detrended Fluctuation Analysis (DFA), which will provide information about the predictability and autocorrelation of the signal, respectively. As a secondary aim, the behavior of the different body segments was compared during the upright stance to understand how joint variability output changes depending on whether the body segments are proximal or distal to the force application point (ground forces).”

Minor points:

Abstract:

  1. Line 18 - the authors should make it clear how a signal can be less predictable while being more auto-correlated (they should add the term "long-term")

Response:

Thank you very much for your comment. We understand the confusion. Overall, experimental results showed that the predictability (entropy) and autocorrelation (DFA) of a signal are moderately related. However, they do not mean the same which could explain why our results are different from the literature.  While entropy quantifies how short-term behaviours are repeated over time, DFA quantifies how further motor patterns depend on previous patterns. We have tried to explain this idea in the reviewed manuscript, and the term long-term has been added along the document for clarification.

Lines 159-160: “FE measures the degree of regularity of the signal output by quantifying how short-term patterns are repeated over time”

Lines 169-171: “On the other hand, DFA evaluates the presence of long-term correlations within the time series by a parameter referred to as the scaling index α (25,29), identifying the extent to which further data are dependent on the previous data (30).”

Introduction:

  1. General - The authors make strong affirmations in terms of "optimal configuration" (Line 29), "very limited DOF" (line 30), "human system always looks for the most stable solution" (lines 35-36), and "motor variability helps the system to find the most successful and stable..." (lines 38-39) (for instance) without much support from the literature. My suggestion is to either cite the ones demonstrating it, or consider the appropriate terms. Considering the last one, I would be careful in citing Davids et al. (2006) as they are not the primary source of evidence backing up the affirmation made.

Response:

The reviewer is right and optimal and stable terms could appear too categorical. From the authors´ point of view “optimal” does not mean “perfect” but “appropriate” according to the specific environmental features coped by the performer in each moment.  For that aim, human motor variability would promote adaptive behaviours by facilitating motor adjustments that refine movement performance during the interaction with changing environment. Nevertheless, we have tried to clarify this idea and update the references in the reviewed version of the manuscript.

Lines 27-30: “The aim of this interact is to achieve a successful motor solution by choosing a suitable degrees of freedom (DoFs) configuration among the multiple and redundant DoFs de-pending on the performer´s level and environment inputs (2).”

  1. Second paragraph: Please unpack the meaning of variability. One can "free" DoFs while being consistent over trials or having a more consistent hand path (for instance).

Response:

Thank you very much for the comment. As we have indicated above, we have modified the introduction section according to the reviewer´s suggestion reducing the “learning” rationale and we have mainly focused on how changes in DOFs are associated with motor variability.

Lines 49-62: “Although there may not be a single pathway of change in the evolving patterns of coordination, Bernstein (8) suggested that "freezing" DoFs strategy would simplify the motor control problem, with the intention of decreasing motor variability. More efficient solutions to the DoFs problem would involve gradual unfreezing or "freeing" of DoFs. Nevertheless, this motor control strategy can paradoxically imply an increase in motor variance in joints coupling but a reduced output variability. Researchers are still investigating how movement synergies actually simplify the problem of movement coordination (8,11-13). To the authors' knowledge, how motor variability changes according to the DOFs chosen for a specific motor task is still unknown.

  1. Line 59 - It is a strong affirmation to claim the human movement system as chaotic. The references cited might not be sufficient for such a claim as, as far as I am aware, they do not demonstrate sensitivity to initial conditions.

Response:

Thank you very much for the comment. This sentence has been changed to be clarified, since, as the reviewer indicates, the affirmation made was not correct. Specifically, the authors have removed the sentence “human movement system as chaotic” and specified that human movement is characterized by nonlinear dynamics, knowing that a chaotic system displays nonlinear behaviours, but not all the nonlinear behaviours are chaotic.

Lines 63-65: It must be taken into account to fully understand the potential relationship between motor variability and the DOFs that the movement dynamics of the biological systems is nonlinear (14–16)”.

  1. Line 59 - "characterized by chaotic dynamics (12-14). That means that the motor variability..." The sentence seems to imply that because the system is chaotic it has deterministic and random parts. A system being chaotic does not mean that it has random and deterministic part. A chaotic system is deterministic by definition.

Response:

Thank you very much for the observation. As it has been indicated in the previous comment, the authors agree with the reviewer. As she/he suggests, a chaotic system exhibits determinism being governed by the simple rules of interaction and extremely sensitive to initial conditions. Thus, the sentence has been rewritten for clarification.

Coffey DS. Self-organization, complexity and chaos: The new biology for medicine. Nat Med. 1998;4(8):882–5.

Lines 63-73: “It must be taken into account to fully understand the potential relationship between motor variability and the DOFs that the movement dynamics of the biological systems is nonlinear (14–16). Thus, even the amount of variability has been deeply related to the exploration of the individual to find a suitable solution (6,17), it does not inform about how movement behavior changes over time (18). Nonlinear tools allow the acquisition of information about the variability structure which allows the quantification of the dynamics of the system's behavior (4,19-21).”

  1. Lines 61-62 - Why is the random part diminished? Is the argument made in terms of Riley & Turvey or Newell & Corcos? They are not empirical papers demonstrating this.

Response:

Thank you very much for the comment. This sentence has been removed, since, as the reviewer indicates, the references used are not empirical papers. In order to improve this part of the introduction and the reason the paper is only focused on the structure of variability additional information has been added before the aim of the study.

  1. Line 70 - "More auto-organized". If I am not mistaked, there is no such thing as more auto-organized (I believe the authors mean "self-organized). Self organization is categorical. Either a system is controlled by (central, external) entity, or the interaction of parts lead to organization (self-organization).

Response:

Thank you very much for the comment. The reviewer is right regarding the meaning of self-organization, but when we used the term “more auto-organized” motor pattern, we meant a behaviour in which a “part” of the system has an increasingly driven role in the final motor output. We have tried to clarify this idea in the reviewed version of the manuscript.

Lines 37-40: Human motor variability would promote these adaptive behaviors by facilitating the exploration of the motor system redundancy (6) and the ability to perform motor adjustments (7), in order to refine movement performance during the interaction with changing environments.”

Methods:

  1. The tasks are poorly motivated.

Response:

Thank you for the comment. As it has been presented above, we have tried to clarify the experiment conditions that we have manipulated and relate it directly with the aims of the study. On the one hand, the manipulations of the DOFs involved in controlling the different tasks’ goals have been done by only modifying the arm behaviour. On the other hand, as a secondary aim, we compared the behaviour of the different joints during a given task (ARelax task) to understand how joint variability output changes depending on whether the joint is proximal or distal to the force application point (ground forces).

This information has been clarified in the introduction and in the method sections.

Lines 90-93: “As a secondary aim, the behavior of the different body segments was compared during the upright stance to understand how joint variability output changes depending on whether the body segments are proximal or distal to the force application point (ground forces).”

Lines 103-106:Participants performed five different simple tasks in one session (Figure 1), in which the kinematic behavior of the distal segment, the arm, was manipulated, from static and constraint conditions to dynamic tasks. Specifically, participants performed three static and two dynamic situations.”

  1. The introduction does not motivate the dynamics/static nature of the tasks being compared.

Response:

Thank you very much for the comment. As we have mention before, the justification of the task conditions applied in the study has been clarified in the introduction and method sections.

Lines 90-93: “As a secondary aim, the behavior of the different body segments was compared during the upright stance to understand how joint variability output changes depending on whether the body segments are proximal or distal to the force application point (ground forces).”

Lines 103-106:Participants performed five different simple tasks in one session (Figure 1), in which the kinematic behavior of the distal segment, the arm, was manipulated, from static and constraint conditions to dynamic tasks. Specifically, participants performed three static and two dynamic situations.”

  1. Lines 121-123: Why were the low frequency fluctuations discarded?

Response:

Thank you very much for the comment. The low-frequency fluctuations were discarded to be able to compare between situations. Since there were some static and some dynamic conditions, the idea was to remove the low-frequency movement displayed from the UpDown and Circle situations. Otherwise, the significant differences in the structure of the variability would be biased by the range of movement of each condition.

  1. Line 139 - a letter "m" is missing in between "vectors of length" and "are less repeatable".

Response:

Thank you very much for the comment. The typo has been fixed. See line 136.

Results:

  1. Line 171: The authors claim more irregularity and auto-correlation. Is it because we are talking about auto-correlation in the long range? That is why it is "contradictory" to irregularity? If this is the case, this must be made clear in a sentence in the methods.

Response:

Thank you very much for your comment. This point has been responded above, regarding the same confusion in the abstract. We have tried to explain this idea in the reviewed manuscript, and the term long-term has been added along the document for clarification.

Lines 159-160: “FE measures the degree of regularity of the signal output by quantifying how short-term patterns are repeated over time”

Lines 169-171: “On the other hand, DFA evaluates the presence of long-term correlations within the time series by a parameter referred to as the scaling index α (25,29), identifying the extent to which further data are dependent on the previous data (30).”

Discussion:

  1. Lines 218-221: The authors should be careful as Bernstein did not refer to synergies in the same sense as Latash puts it currently (in terms of redundant, non-redudant spaces). My suggestion is for the authors to specify clearly what is meant here.

Response:

Thank you very much for the observation. Certainly, Bernstein and Latash talk about the concept of synergies but from different perspectives. As we meant to the functional coupling between degrees of freedom as Bernsteins suggested, we have removed in this sentence the reference to Latash (in terms of a neural organization that ensures co-variation among elemental variables along time or across repetitive attempts at a task). See line 290.

Reviewer 2 Report

In this paper authors described the relationship between the human motor noise and degrees of freedom involved in movement control. This paper is well written and well presented. I have following suggestions/questions:

1. On page 3 (instruments and procedures) authors mentioned that they used electromagnetic tracking sensor. How was data processed? Did they use any filtering to remove noise? 

2. Electromagnetic sensors are effected by external electromagnetic interference. Did authors considered this (i.e., all participants participated in the same room under same conditions)?

3. Table 1: double spacing is being used whereas rest of document is not. Also, please use zero on the left of the decimal point (i.e., 0.427 not .427)

4. Figure 3 (left) is very hard to interpret. Revise this and may be share data in the table form 

Author Response

We thank the editor and the reviewers for the opportunity to enhance the quality of the manuscript. Below is each issue raised by the reviewers (in bold font) followed by the answers. The changes in the manuscript have been marked up using the Track Changes:

REVIEWER 2

Minor points:

  1. On page 3 (instruments and procedures) authors mentioned that they used electromagnetic tracking sensor. How was data processed? Did they use any filtering to remove noise?

Response:

Thank you very much for the comment. As the reviewer suggested, the Polhemus Liberty® tracking system performs a data process to remove noise signal. Specifically, the data were filtered with a single-pole low-pass filter with an adaptive pole location. The pre-set filtering parameters were: sensitivity=0.02; boundary (FLow)=0.02: boundary (FHigh)=0.8; Max transition rate=0.95. This information has been added to the Data analysis and reduction section.

Lines 141-144: “Data were already filtered by the Polhemus Liberty® tracking system with a sin-gle-pole low-pass filter with an adaptive pole location. The pre-set filtering parameters were: sensitivity=0.02; boundary (FLow)=0.02: boundary (FHigh)=0.8; Max transition rate=0.95.”

  1. Electromagnetic sensors are affected by external electromagnetic interference. Did authors considered this (i.e., all participants participated in the same room under same conditions)?

Response:

Thank you very much for your comment. As the reviewer indicates, the electromagnetic sensors used are sensitive to electromagnetic interference. That is why all the participants performed the experimental protocol under the same conditions, being placed in the same space of the laboratory, which was calibrated.

  1. Table 1: double spacing is being used whereas rest of document is not. Also, please use zero on the left of the decimal point (i.e., 0.427 not .427)

Response:

Thank you very much for your comment.  The spacing of Table 1 has been changed to simple spacing, and the zeros on the left of the decimal point have been added.

  1. Figure 3 (left) is very hard to interpret. Revise this and may be share data in the table form.

Response:

Thank you very much for your suggestion. The Standard Deviation information has been removed from Figure 2 y 3. The mean and SD of this variable for all the situations can be checked in Table 1, and the pair comparisons have been included in the text.

Lines 203-207: “Specifically, in the situations with voluntary movement, participants also showed significantly higher SD and lower FE values than in relaxed situations (HoS and ARelax situations) (p < .001). In addition, in the situations in which the muscular contractions were dynamic (UpDown and Circle), the participants presented significantly higher SD and lower FE values (p < .01).”

Lines 232-233: “The amount of variability was significantly lower in the hip sensor than in the shoulder and hand sensor (p < .001)”

Round 2

Reviewer 1 Report

The authors addressed my comments. I believe it is in good shape for publication.